# Management of Glyphosate-Resistant Weeds in Mexican Citrus Groves: Chemical Alternatives and Economic Viability

**DOI:** 10.3390/plants8090325

**Published:** 2019-09-04

**Authors:** Ricardo Alcántara-de la Cruz, Pablo Alfredo Domínguez-Martínez, Hellen Martins da Silveira, Hugo Enrique Cruz-Hipólito, Candelario Palma-Bautista, José Guadalupe Vázquez-García, José Alfredo Domínguez-Valenzuela, Rafael De Prado

**Affiliations:** 1Departamento de Química, Universidade Federal de São Carlos, São Carlos 13565-905, Brazil; 2National Institute of Forestry, Agriculture and Livestock Research (INIFAP)-Valle del Guadiana Experimental Field, Durango 34170, Mexico; 3Departamento de Fitotecnia, Universidade Federal de Viçosa, Viçosa 36570-900, Brazil; 4Bayer Crop Science Mexico, Mexico 11520, Mexico; 5Department of Agricultural Chemistry and Edaphology, University of Cordoba, 14071 Cordoba, Spain (C.P.-B.) (J.G.V.-G.) (R.D.P.); 6Department of Agricultural Parasitology, Chapingo Autonomous University, Texcoco 56230, Mexico

**Keywords:** *Citrus latifolia*, hairy beggarticks, integrated weed management, junglerice, tropical sprangletop

## Abstract

Glyphosate is a cheap herbicide that has been used to control a wide range of weeds (4–6 times/year) in citrus groves of the Gulf of Mexico; however, its excessive use has selected for glyphosate-resistant weeds. We evaluated the efficacy and economic viability of 13 herbicide treatments (glyphosate combined with PRE- and/or POST-emergence herbicides and other alternative treatments), applied in tank-mixture or sequence, to control glyphosate-resistant weeds in two Persian lime groves (referred to as SM-I and SM-II) of the municipality of Acateno, Puebla, during two years (2014 and 2015). The SM-I and SM-II fields had 243 and 346 weeds/m^2^, respectively, composed mainly of *Bidens pilosa* and *Leptochloa virgata*. *Echinochloa colona* was also frequent in SM-II. The glyphosate alone treatments (1080, 1440, or 1800 g ae ha^−1^) presented control levels of the total weed population ranging from 64% to 85% at 15, 30, and 45 d after treatment (DAT) in both fields. Mixtures of glyphosate with grass herbicides such as fluazifop-p-butyl, sethoxydim, and clethodim efficiently controlled *E. colona* and *L. virgata,* but favored the regrowth of *B. pilosa.* The sequential applications of glyphosate + (bromacil + diuron) and glufosinate + oxyfluorfen controlled more than 85% the total weed community for more than 75 days. However, these treatments were between 360% and 390% more expensive (1.79 and 1.89 $/day ha^−1^ of satisfactory weed control, respectively), compared to the representative treatment (glyphosate 1080 g ae ha^−1^ = USD $29.0 ha^−1^). In practical and economic terms, glufosinate alone was the best treatment controlling glyphosate resistant weeds maintaining control levels >80% for at least 60 DAT ($1.35/day ha^−1^). The rest of the treatments, applied in tank-mix or in sequence with glyphosate, had similar or lower control levels (~70%) than glyphosate at 1080 g ae ha^−1^. The adoption of glufosiante alone, glufosinate + oxyfluorfen or glyphosate + (bromacil + diuron) must consider the cost of satisfactory weed control per day, the period of weed control, as well as other factors associated with production costs to obtain an integrated weed management in the short and long term.

## 1. Introduction

Citriculture is an important activity in Mexico occupying ~40% of the total area devoted to fruticulture [1]. The state of Veracruz is the biggest producer of citrus fruits with ~225,000 ha, and the orange (*Citrus sinensis*) occupies the largest crop area (55%); however, the Persian lime (*C. latifolia*), with ~47,000 ha [2], makes the highest economic contribution [3]. In this way, Persian lime plantations receive more care than other citrus crops from small and large growers [4], such as fertilization (foliar and soil), prunings, control of diseases (*Colletotrichum gloeosporioides*), pests (*Diaphorina citri* and *Phyllocnistis citrella*), and weeds (chemical and mechanical) [5,6,7].

Weed competition require special attention in young citrus trees and during blooming and fruit setting [8], but in older plantations, the impacts of weeds are also indirect, mainly by limiting crop management [9], in addition to the fact that weeds can be hosts for pests and diseases [6]. Thus, weed management in the citrus-producing region of the Gulf of Mexico, by combining chemical and non-chemical (manual or mechanical mowing) methods, is carried out four to six times a year [7,9,10], representing ~8–12% of production costs ha^−1^ [11].

Glyphosate is a systemic and non-residual and post-emergence (POST) herbicide that controls a wide range of weeds [12], making it preferred by the Mexican citrus growers [7,13]. The doses recommended by the manufacturers of this herbicide range from 700 to 2100 g ae ha^−1^ [14], according to the weed species, phenological stage, and infestation level. Due to the frequent glyphosate applications, most Mexican citrus growers have widely adopted and applied doses ranging from 720 to 1080 g ae ha^−1^ for up to 15 years [9]. The high dependence of glyphosate-based herbicides has led to the selection of resistant populations of *Bidens pilosa* [15], *Eleusine indica* [4], *Leptochloa virgata* [7], and *Parthenium hysterophorus* [13] between 2010 and 2016 in the citrus-producing region of the states of Puebla and Veracruz, Gulf of Mexico. 

Despite the evident loss of glyphosate efficacy in controlling weeds, Mexican citrus growers continue using this herbicide for its low cost, which makes it necessary to look for weed management alternatives that help to extend the useful life of this herbicide [12]. Glufosinate and paraquat, POST, broad-spectrum, and non-residual herbicides like glyphosate are also used in citrus groves [7]. However, these herbicides have short control periods forcing growers to make more applications, which is more expensive than with glyphosate alone. In addition, when glufosinate or paraquat are applied in late POST in order to reduce the number of herbicide applications, weed control is poor (personal communication with growers). Management strategies with pre-emergence (PRE) and early POST herbicides could reduce the selection pressure exerted by glyphosate on weeds that are difficult to control [16], as well as production costs [11].

In this work, we evaluated the efficacy and economic viability of 13 herbicide treatments (glyphosate combined with PRE and/or POST herbicides and other alternative treatments), applied in tank-mixture or sequence (Table 1), to control weeds, including glyphosate-resistant species in the citrus-producing region of the Gulf of Mexico.

## 2. Results

### 2.1. Initial Weed Density

The average density of weeds was 242.9 and 345.6 plants m^2^ in the SM-I and SM-II fields, respectively. Weed community was composed mainly of *B. pilosa* and *L. virgata* in both fields. In addition, *Echinochloa colona* was frequent in SM-II. Density of weeds showed no differences between years and *B. pilosa* presented the highest density (Table 2). Species such as *Amaranthus viridis*, *Cynodon nlemfuensis*, *Digitaria sanguinalis, Eleusine indica,* and *Parthenium hysterophorus* were sporadic in SM-I, and *E. indica* and *Rottboellia cochinchinesis* in SM-II. Due to the low density of these weeds, they were not considered for the analysis of herbicide control per species.

### 2.2. Total Control of Weeds 

The glyphosate treatments of 1080, 1440, and 1800 g ae ha^−1^ presented similar control levels at 15, 30, and 45 days after treatment (DAT) in both SM-I and SM-II fields ranging from 64% to 85%. Weed control with 1080 g ae ha^−1^ of glyphosate was ~10–20% lower in relation to the other two-glyphosate treatments at the 60 and 75 DAT. Most of the herbicides, applied in tank-mix or in sequence with glyphosate, had similar control levels (~70%) than the lowest dose of glyphosate (1080 g ae ha^−1^) at 15 DAT, except clethodim and oxadiazon in both fields, and fluazifop-p-butyl, oxyfluorfen + fluazifop-p-butyl (in sequence), and sethoxydim in SM-II that showed lower control level than glyphosate alone. Acetochlor in SM-II and fluazifop-p-butyl, sethoxydim, and clethodim in tank mixture, and fluazifop-p-butyl and oxadiazon in sequential application in SM-II, showed greater control at 30 DAT than at 15 DAT. As of this period, the control of these herbicides was similar or less than the control obtained with glyphosate at 1080 g ae ha^−1^, except the sequential application of bromacil + diuron. The control level of the latter treatment increased from 30 DAT up to 90% until the end of the experiments. Glufosinate and glufosinate + oxyfluorfen (in sequence) showed the highest levels of control (>95%) at 15 and 30 DAT. The last treatment showed a control level above 90% at 75 DAT, while glufosinate alone decreased to 77%. Paraquat + diuron had control levels above 85% at 15 and 30 DAT but decreased to 48% at 75 DAT (Table 3).

### 2.3. Control of Bidens Pilosa 

The control of *B. pilosa* with the different treatments was more heterogeneous in SM-I at 30 DAT than in SM-II. None of the three-glyphosate treatments showed satisfactory levels of control. Glufosinate alone and the sequential applications of glufosinate + oxyfluorfen and glyphosate + (bromacil + diuron) presented the best control levels of *B. pilosa* in both fields at 30 and 75 DAT. Paraquat + diuron presented a control >80% at 30 DAT but decreased to 50% at 75 DAT. The other herbicides, applied in tank-mix or in sequence with glyphosate, had similar or lower control than any glyphosate treatment alone. As expected, graminicides such as fluazifop-p-butyl, sethoxydim, and clethodim did not contribute to the control of *B. pilosa* (Figure 1).

### 2.4. Control of Leptochloa Virgata

Most of the treatments controlled *L. virgata* by 85% or higher in both fields at 30 DAT, except glyphosate alone (1080, 1440, and 1800 g ae ha^−1^) and the sequential application with acetochlor in SM-II (control ~60%). At 75 DAT, tank-mixtures of glyphosate with fluazifop-p-butyl, oxyfluorfen + fluazifop-p-butyl (in sequence), and clethodim had control levels of ~90% in SM-I. Oxifluorfen, applied in sequence, extended the control of *L. virgata* (~80%) with glufosinate. Paraquat + diuron and the sequential applications of glyphosate with oxadiazon and bromacil + diuron maintained control levels of 76–82%. At SM-II, all herbicides, applied in tank-mixture or in sequence with glyphosate, showed greater control of *L. virgata* compared to the glyphosate alone treatments. Oxifluorfen + fluazifop-p-butyl, sethoxydim, oxadiazon, and bromacil + diuron contributed to maintaining control levels >85% at 75 DAT. Treatments that did not include glyphosate had similar control levels (glufosinate) or lower (glufosinate + oxyfluorfen and paraquat + diuron) than the previous treatments. However, these levels of control were similar to those observed in SM-I in the same period (Figure 2).

### 2.5. Control of Echinochloa Colona

*Echinochoa colona* occurred at high density in the field SM-II, but it was controlled by most of the treatments, including the glyphosate ones, in more than 90% at 75 DAT. Glufosinate and the mixture of paraquat + diuron showed the lowest control levels (~75%) due to low residuality (Figure 3).

### 2.6. Cost of Glyphosate Resistance Management

Analyzing the cost of treatments compared to the representative treatment most used by producers (glyphosate at 1080 g ae ha-1 = USD $29.0 ha^−1^), with exception of the paraquat + diuron that was 8% cheaper, all treatments were between 33 and 391% more expensive. Taking into account the duration of the experiments (75 days), the cheapest daily cost of weed control was $ 0.36/day for the cheapest treatment (duron + paraquat), and glufosinate + oxyfluorfen and glyphosate + (bromacil + diuron) were apparently the most expensive treatments (1.79 and $ 1.89 $/day ha^−1^, respectively) (Table 4). The minimum satisfactoty and accdeptable level of weed control for an herbicide treatment is 80%, according to the European Weed Research Society [17]. Based on this criterion, several treatments did not reach this level of weed control since 15 DAT; mainly in the San Manuel I field (Table 2). Thus, the treatments that presented most expensive cost of satisfactory weed control per day were glyphosate (1800 g ae ha^−1^, $3.22/day ha^−1^) in San Manuel I and glyphosate + fluazifop-p-butyl in San Manuel II ($3.36/day ha^−1^), since they only maintained control ≥ 80% for 15 and 30 DAT, respectively. Glufosinate + oxyfluorfen and glyphosate + (bromacil + diuron) presented control levels ≥ 80% for 75 DAT in both fields, i.e., they were almost half-cheaper than the previous two treatments. However, the best satisfactory cost of weed control per day (0.95 and 1.08 $/day ha^−1^) was achieved with glyphosate + acetochlor (only in San Manuel II) and glufosinate (in both fields), mainly by maintaining weed control ≥ 80% at least 60 DAT (Table 4).

## 3. Discussion

Weed community was mainly composed of *B. pilosa*, *E. colona*, and *L. virgata* in both SM-I and SM-II fields. Weed control in citrus groves of the Gulf of Mexico has been based mainly on glyphosate-based herbicides [4,7,13,15,18]. This almost exclusive dependence exerted a strong selection pressure favoring the establishment of these species in the Persian lime groves of the San Manuel Farm. Natural weeds are megadiverse, composed of both dicot and monocotyledons [19,20]; however, most cropping practices reduce plant diversity favoring few weed species [21], as observed in the SM-II field that showed a lower weed diversity than SM-I field, but higher density of plants. 

The glyphosate-based treatments with increasing doses did not increase the total control of weeds, mainly, of the dominant species *B. pilosa* and *L. virgata*. Glyphosate resistance of *L. virgata* in citrus groves from Martinez de la Torre and Cuitláhuac, state of Veracruz, was confirmed in 2010 [7,22], municipalities that are at least 450 km apart. Some of these populations selected glyphosate resistance independently, but the majority have a common resistance selection origin that was spread throughout the citrus-producing region of the Gulf of Mexico to both short and long distances mainly by dispersing its seeds by people, tractors, and machinery used for the cultural deals [22]. Considering that the municipality of Acateno is located ~25 km from Martinez de la Torre, it was expected that glyphosate-resistant populations of *L. virgata* found in the San Manuel Farm, as well as in other farms around the region. In addition, there could also be selection of glyphosate resistance of weeds in situ [22], because weed management strategies based on herbicides lead to an eventual loss of control [23], and consequently, the selection for weeds resistant to herbicides [24].

The continuous use of glyphosate in citrus groves of the San Manuel farm, in addition to consolidating the glyphosate resistance of *L. virgata*, also selected for resistance in *B. pilosa*, as later corroborated by characterizing representative populations (R1 and R2 collected in SM-II and SM-I fields, respectively [15]). Some grass-controlling herbicides (fluazifop-p-butyl, sethoxydim, and clethodim), which efficiently controlled *L. virgata* and *E. colona*, reduced the efficacy of glyphosate to control *B. pilosa*, i.e., were apparently antagonistic. However, the low levels in controlling this species were perhaps due to its resistance to glyphosate, because once grasses were controlled by the grass herbicides, uncontrolled *B. pilosa* plants regrowth reached high levels of coverage, which was reflected in the low control percentages of total weeds of these treatments. This shows that improper implementation of a weed management strategy may indirectly favor the selection of herbicide resistance in another weed(s) [25]. This situation is worrisome because infestations of *B. pilosa* may impede future manual activities of crop management such as pruning and harvesting (Figure 4), since its achenes have three barbed spines that easily attach to clothing or fur [26]. Bromacil + diuron applied 15 DAT following glyphosate application controlled *B. pilosa*, demonstrating its contribution of bromacil + diuron in the control of weeds. Thus, other mixtures of glyphosate with herbicides such as 2,4-D, dicamba, or picloram, with good efficacy on dicots [27], could also contribute to controlling *B. pilosa* in autumn–winter period when it reaches its highest population density, as well as other broadleaf weeds, when trees are not blooming or setting fruits. 

The poor control of *L. virgata* with the glyphosate-based treatments with acetochlor and oxadiazon in both the SM-I and SM-II fields was due to regrowth of plants. In situations where glyphosate poorly controls resistant weeds, the use of residual herbicides was proposed to control grasses that come from seeds [28]. The mixture of glyphosate with contact or residual herbicides can improve weed control in citrus plantations in Florida, USA, and at the same time reduce the selection of resistant populations and extend the weed control period [16]. Tank-mix of glufosinate + indaziflam controlled glyphosate-resistant *L. virgata* populations in Persian lime and orange groves from Veracruz, Mexico, up to 90 days due to the residually of the second herbicide [7]. However, most Mexican citrus growers have not adopted the practice of using PRE and residual herbicides for effective and timely control of weeds. In addition, growers are reluctant to adopt non-chemical weed control methods [29,30]. Thus, the misuse of herbicides (late application, subdose, or overdose) has selected for resistant weeds, which increased yield losses and production costs [30].

The good control of *E. colona*, a weed species in high plant density in the SM-II field, and the other sporadic weeds were also susceptible to glyphosate. However, the high occurrence rates of *B. pilosa* and *L. virgata* resistant to glyphosate make it necessary to seek weed management strategies, including non-chemical methods, without increasing production costs.

Without the occurrence of glyphosate resistant weeds, the cost of satisfactory weed control per day would be $ 0.38/day ha^−1^ (1800 g ae ha^−1^ glyphosate). However, due to the occurrence of *B. pilosa* and *L. virgata* populations resistant to this herbicide, no glyphosate treatment alone showed control above 80%, classified as satisfactory or good [17], for more than 30 DAT. Glufosinate + oxifluorfen and glyphosate + (bromacil + diuron) maintained this control level up to 75 DAT in both fields, but in global terms, they were the more expensive treatments (1.79 and 1.89 $/day ha^−1^, respectively). The cost of an herbicide treatment, without considering the application cost, is a good indicator that can help in selecting a chemical treatment [29,31], but this cost is relevant when the choice considers the period of control. In this sense, mixtures of glyphosate with sethoxydim, cletodim or oxadiazon were not efficient to control weeds in practical and economic terms. However, the best treatment was glufosinate that maintained an acceptable weed control for 60 DAT in San Manuel I ($1.35/day ha^−1^) and 75 in San Manuel II ($1.08/day ha^−1^). This represents at least one application operation less in comparison to the other treatments that presented control > 80% for up to 30 DAT. Therefore, the choice of a relative more expensive herbicide treatments, but that provide an acceptable control of glyphosate resistant weeds for longer in Persian lime orchards could result in fewer herbicide applications, reducing the cost of this and other crop management tasks [32], such as pruning and harvest. Integrated herbicide-resistant weed management programs reduce profits in the first year of its implementation, but profits increase in the second and subsequent years [33,34], since once the seed bank decreases, less expensive and persistent active ingredients can be used [23,35]. This could improve the yield and quality of fruits, increasing the return, since weed management is a key component to sustaining productivity [33,34]. Therefore, the adoption of weed management strategies, either chemical, not chemical, or combined, must consider other factors associated with production costs rather than the individual cost of a determined control method, in order to obtain an integrated weed management in the short and long term [29,30].

## 4. Materials and Methods 

### 4.1. Local Data and Experimental Design

Field trials was carried out in two Persian lime groves, of the “San Manuel” Farm (20.10° N, 97.16° W), in Acateno, Puebla, referred to as SM-I and SM-II (at 128 and 137 m above sea level, respectively), which had received glyphosate applications of 720–1080 g ae ha^−1^ between 3 and 5 times a year from 2006 to 2014. The groves were at least 800 m apart, and Persian lime trees were 3 years old and arranged at 3 × 5 m between trees and rows, respectively. Thirteen herbicide treatments plus an untreated control (Table 1) were distributed in each field in a randomized block design with three replications. Each plot consisted of a 6 × 5 m area including two trees. A representative soil sample from sub random samples (0–10 cm) taken in both fields was analyzed. The sandy clay loam (8% clay, 64% sand, and 28% silt) presented pH = 8.06, 3.3% organic matter; 10.4 and 15.1 mg kg^−1^ of nitrates and P; and 0.01, 24.8, 0.93, 0.03, and 0.39 Cmol kg^−1^ of Ca, Mg, Na, and P, respectively. Field trials were carried out in 2014 and repeated in 2015 and the climatic conditions, presented during the execution of the experiments that were recovered from the National Meteorological System of Mexico, there are outlined in the Figure 5.

### 4.2. Herbicide Application 

A mechanical mowing of weeds was carried out in the two experimental fields three weeks before application of herbicide treatments, following the crop management practices of the Mexican citrus growers. The applications were made on 10–15 cm tall weeds using a motorized manual spray backpack (Swissmex 425 SW), equipped with pressure regulator and an AI11002 nozzle, calibrated to spray 277 L ha^−1^ at 40 psi. The water pH of the glufosinate and glyphosate treatments was adjusted to 5.5 with the acidifying-buffing adjuvant pHase1^®^ (Arysta LifeSciences México, Saltillo, México). The PRE herbicides of the treatments 5, 6, 11, 12, and 13 were applied 15 days after application of POST herbicides. Table 5 summarizes the action dates of field activities during the experiments. 

### 4.3. Evaluated Variables

One day before the herbicide applications, weeds were identified and counted using a square frame of 0.25 m^2^, randomly placed twice in each experimental unit. These data were expressed in number of plants per m^2^. The visual control of weeds total and by species was conducted at 15, 30, 45, 60, and 75 days after treatment (DAT). The evaluation at 75 DAT was included considering that PRE herbicides would have 60 days of activity. The control percentage of total weeds or a determined species was estimated with the equation X = [(A − B)/A] × 100 [37], which compares the average coverage of weeds in plot (B) vs. the weed population in the control plot. The parameters of equation represent: X the control percentage of total weed or a determined species; A the coverage percentage of total weed or a determined species in the control plot, and B the coverage percentage of total weed or a determined species in the treatment evaluated.

Cost of herbicide treatments per ha was recorded to estimate the relative daily cost of weed control considering the evaluation period (75 days), as well as the percentage in cost increase of each treatment in relation to the representative treatment (1080 g ae ha^−1^ glyphosate = USD $ 29.0) most used by farmers. The herbicides treatments that presented weed control >80% as well as DAT that maintained that control level were identified to estimate the cost of satisfactory weed control per day.

### 4.4. Statistical Analysis

Percentage data were transformed to arcsine and, then, the model assumptions of normal distribution of errors and homogeneous variance were graphically inspected. The variance stability tests of control percentage data showed no differences for both cropping seasons, and data were pooled and subjected to ANOVA. Significant differences between means were analyzed using the Tukey’s test at the 0.05 probability level. Statistical analysis was conducted using the Statistix 9.0 software (Analytical Software, Tallahassee, FL, USA).

## 5. Conclusions

Glyphosate alone, even at high doses, had difficulty controlling *B. pilosa* and *L. virgata*. Glyphosate applied in mixture with POST grass herbicides controlled *E. colona* and *L. virgata*, but favored the regrowth of glyphosate-resistant *B. pilosa* plants. Glufosinate alone, glufosiante + oxyfluorfen and glyphosate + (bromacil + diuron) controlled t weeds above 80% for at least 60 d, but they were also the most expensive herbicides. In practical and economic terms, glufosinate alone was the best treatment controlling glyphosate resistant weeds in Persian lime orchards. Therefore, the choice or not of these treatments must consider the cost of satisfactory weed control per day, the period of weed control, as well as other factors associated with production costs, and not only the relative cost of a certain treatment. Integrated weed management programs focused on reducing herbicide dependence that favor weed diversity in the short and long term, as well as facilitating other cultural tasks, should be implemented in the citrus-producing region of the Gulf of Mexico.

## Figures and Tables

**Figure 1 plants-08-00325-f001:**
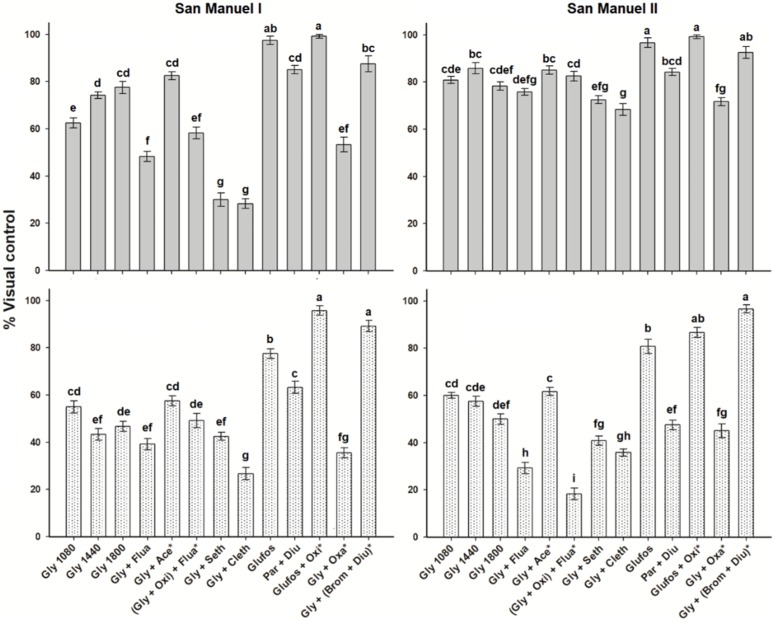
Control of *Bidens pilosa* in two Persian lime groves of the “San Manuel” Farm, Puebla, Mexico, at 30 (gray bars) and 75 (dotted bars) d after treatment. Same letter within a subfigure showed no differences between treatments by the Tukey test (*P* > 0.05). Vertical bars ± standard error from combined data of field trials carried out in 2014 and 2015 (*n* = 6).

**Figure 2 plants-08-00325-f002:**
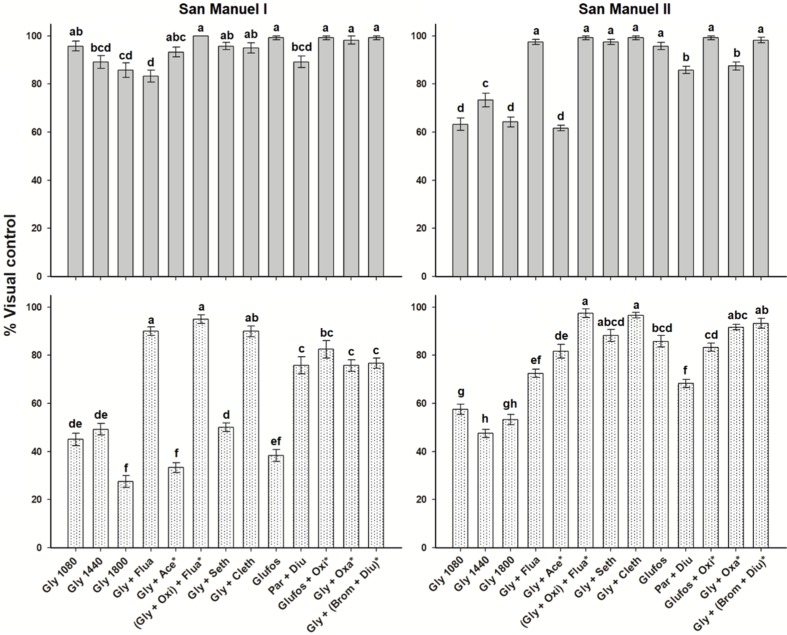
Control of *Leptoclhoa virgata* in two Persian lime groves of the “San Manuel” Farm, Puebla, Mexico, at 30 (gray bars) and 75 (dotted bars) d after treatment. Same letter within a subfigure showed no differences between treatments by the Tukey test (*P* > 0.05). Vertical bars ± standard error from combined data of field trials carried out in 2014 and 2015 (*n* = 6).

**Figure 3 plants-08-00325-f003:**
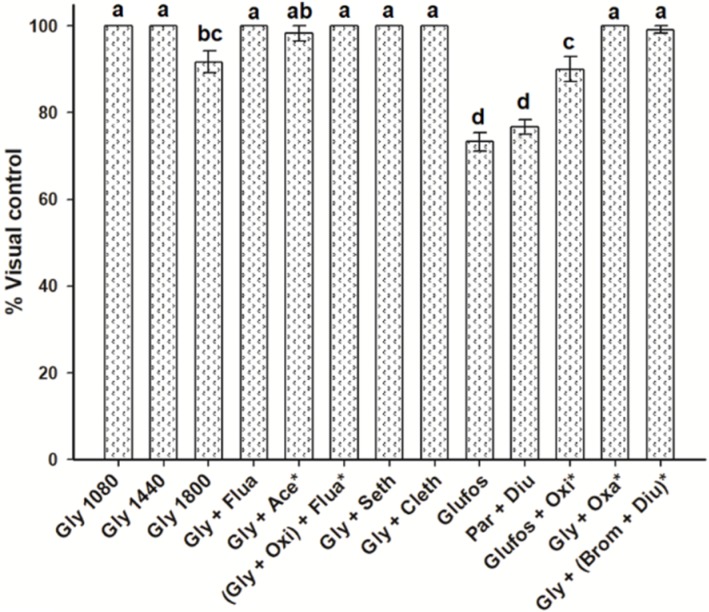
Control of *Echinochloa colona* in the Persian lime grove “San Manuel II” of the “San Manuel” Farm, Puebla, Mexico, at 75 days after treatment. Same letter shows no differences between treatments by the Tukey test (*P* > 0.05). Vertical bars ± standard error from combined data of field trials carried out in 2014 and 2015 (*n* = 6).

**Figure 4 plants-08-00325-f004:**
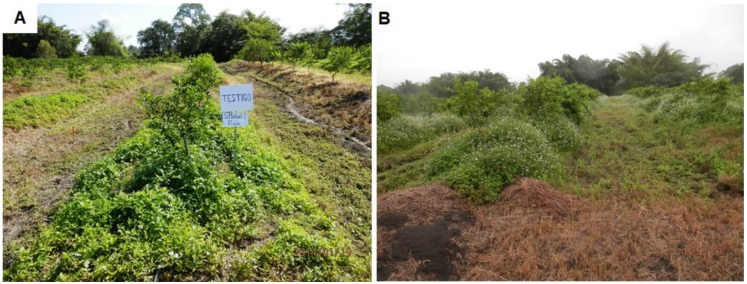
Persian lime rows infested with *B. pilosa* in “San Manuel I”, Acateno, Puebla, in 2014. (**A**) Untreated control plot 10 DAT, and (**B**) random rows at 30 DAT. Persian lime trees were 3 years old.

**Figure 5 plants-08-00325-f005:**
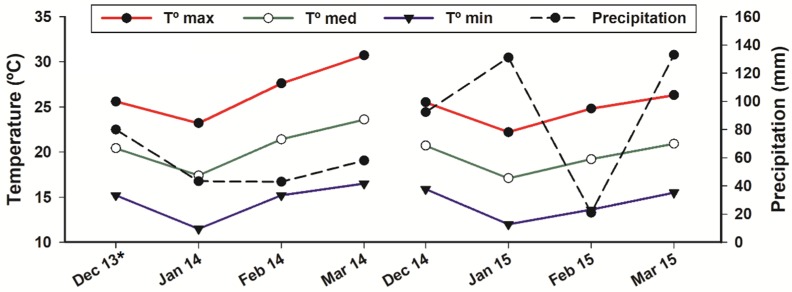
Accumulated precipitation (mm) and average monthly temperatures (°C) in Acateno, Puebla, Mexico from December 2013 to March 2014 and December 2014 to March 2015 (Source SMN, 2019 [36]). * Data are the historical average of the parameters presented since there is no data available for this month.

**Table 1 plants-08-00325-t001:** Herbicides (Treatments), mechanism of action (MOA), field rates in g ai or ea ha^−1^ (Rate), liters of commercial herbicide ha^−1^ (liters), application time (Time) of pre- (PRE) and post-emergence (POST) herbicides for the weed control in two Persian lime groves of the “San Manuel” Farm, Puebla, Mexico, and cost of each treatment ha^−1^ (USD).

	Treatments ^1^	MOA ^2^	Rate	Liters	Time	USD ^3^
-	Control	-	-	-	-	-
1	Gly 1080	EPSPS	1080	3	POST	29.0
2	Gly 1440	EPSPS	1440	4	POST	38.7
3	Gly 1800	EPSPS	1800	5	POST	48.4
4	Gly + Flua	EPSPS + ACCase	1080 + 250	3 + 2	POST	100.9
5	Gly + Ace ^†^	EPSPS + Mitosis	1080 + 1678	3 + 2	POST+PRE	57.3
6	(Gly + Oxi) + Flua ^†^	(EPSPS+PPO) + ACCase	(1080 + 480) + 250	(3 + 2) + 2	POST+PRE	97.4
7	Gly + Seth	EPSPS + ACCase	1080 + 368	3 + 2	POST	82.0
8	Gly + Cleth	EPSPS + ACCase	1080 + 236	3 + 2	POST	70.0
9	Glufos	GS	450	2	POST	54.1
10	Par + Diu	PSI + PSII	400 + 200	2	POST	26.8
11	Glufos + Oxi ^†^	GS + PPO	420 + 480	2 + 2	POST + PRE	134.6
12	Gly + Oxa ^†^	GS + PPO	1080 + 1000	4 + 3	POST + PRE	112.0
13	Gly + (Brom + Diu) ^†^	EPSPS + (PSII + PSII)	1080 + (1200 + 1200)	3 + 3	POST + PRE	142.4

^1^ Gly = Faena^®^ Fuerte 360 (SC, 35.6% glyphosate w/v); Flua = Fusilade BIW^®^ (EC, 12.5% fluazifop-p-butyl w/v); Ace = Harness^®^ EC (EC, 60% acetochlor w/v); Oxi = Goal^®^ 2XL (EC, 22.3% oxifluorfen w/v); Seth = Poast^®^ (CL, 18.4% sethoxydim w/v); Cleth = Select^®^ Ultra (CE, 12.5% clethodim w/v); Glufos = Finale^®^ (CE, 15% glufosinate w/v); Par + Diu = Gramocil^®^ (SC, 20 + 10% paraquat + diuron w/v); Oxa= Ronstar^®^ 25CE (CE, 24.4% oxadiazon w/v); and Brom + Diu = Krovar^®^ (WG, 40 + 40% bromacil + diuron w/w). *Mention of trade names in this publication is solely for providing specific information and does not imply their recommendation.*
^2^ Mechanism of action: Inhibitors of enolpyruvyl shikimate-3-phosphate synthase (EPSPS), acetyl-CoA carboxylase (ACCase), mitosis, protoporphyrinogen oxidase (PPO), glutamine synthetase (GS), photosystem I (PSI) and II (PSII). ^3^ Average exchange rate of the Mexican peso (MNX) to US dollar (USD) corresponding to January 2014 (13.20 = 1.0) and January 2015 (14.67 = 1.0), respectively. ^†^ Treatments applied in sequence 15 days after the first application.

**Table 2 plants-08-00325-t002:** Initial weed density (plants m^2^) in two Persian lime groves of the “San Manuel” Farm, Puebla, Mexico.

Species	San Manuel I		San Manuel II	
	2014	2015	2014	2015
*B. pilosa*	114.8 ± 4.6	117.3 ± 3.7	195.9 ± 8.6	182.5 ± 6.3
*L. virgata*	98.7 ± 4.7	104.8 ± 3.4	49.8 ± 3.0	58.7 ± 4.6
*E. colona*	7.3 ± 3.2	11.4 ± 2.08	96.7 ± 5.9	87.3 ± 4.1
Other weeds	19.6 ± 2.6	15.6 ± 1.7	8.0 ± 2.1	12.4 ± 3.8
Total	237.7	248.1	350.3	340.9

±Standard error of the mean (*n* = 28).

**Table 3 plants-08-00325-t003:** Total weed control percentage with pre- and post-emergence herbicides in two Persian lime groves of the “San Manuel” Farm, Puebla, Mexico from 15 to 75 days after treatment (DAT). Visual control was measured as 0 = no control and 100 = plant death.

Treatment ^1^	15 DAT	30 DAT	45 DAT	60 DAT	75 DAT
San Manuel I					
Control	-	-	-	-	-
Gly 1080	74.2 ± 2.4 c	64.2 ± 1.5 e	68.3 ± 1.7 d	48.3 ± 2.8 ef	41.7 ± 2.5 ef
Gly 1440	72.5 ± 3.1 c	78.3 ± 1.1 cd	73.3 ± 2.1 cd	64.2 ± 3.3 cd	49.2 ± 2.0 de
Gly 1800	84.2 ± 2.0 b	76.7 ± 2.1 d	78.3 ±1.1 c	62.5 ± 2.8 cd	55.0 ± 1.8 cd
Gly + Flua	68.3 ± 1.7 c	51.7 ± 1.7 f	53.3 ± 3.1 ef	41.7 ± 2.5 f	35.8 ± 2.0 f
Gly + Ace ^†^	69.2 ± 2.4 c	84.2 ± 2.0 bcd	71.7 ± 1.7 cd	59.2 ± 3.0 de	63.3 ± 2.1 c
(Gly + Oxi) + Flua ^†^	71.7 ± 2.5 c	64.2 ± 1.5 e	70.8 ± 1.5 cd	64.2 ± 2.4 cd	52.5 ± 1.7 d
Gly + Seth	73.3 ± 2.1 c	37.5 ± 1.7 g	58.3 ± 2.1 e	55.0 ± 2.2 de	45.0 ± 1.8 def
Gly + Cleth	66.7 ± 1.7 c	34.2 ± 2.7 g	46.7 ± 2.5 f	42.5 ± 2.1 f	20.0 ± 2.9 g
Glufos	100.0 ± 0 a	97.5 ± 1.1 a	88.3 ± 2.1 b	85.8 ± 1.5 b	77.5 ± 1.1 b
Par + Diu	86.7 ± 1.1 b	86.7 ± 2.1 bc	74.2 ± 0.8 cd	70.8 ± 0.8 cd	54.2 ± 2.4 cd
Glufos + Oxi ^†^	100.0 ± 0 a	99.2 ± 0.8 a	97.5 ± 1.8 a	96.3 ± 1.7 a	91.7 ± 1.7 a
Gly + Oxa ^†^	65.0 ± 1.3 c	68.3 ± 2.5 de	59.2 ± 1.5 e	38.3 ± 2.5 f	35.8 ± 2.4 f
Gly + (Brom + Diu) ^†^	70.8 ± 2.7 c	88.3 ± 1.7 b	91.7 ± 2.5 ab	92.5 ± 2.5 ab	89.2 ± 2.7 a
San Manuel II					
Control	-	-	-		-
Gly 1080	66.7 ± 2.5 cd	80.8 ± 4.0 de	69.2 ± 1.5 bc	61.7 ± 3.3 cd	43.3 ± 2.1 ef
Gly 1440	76.7 ± 2.1 b	80.8 ± 1.5 de	69.2 ± 2.0 bc	70.8 ± 2.0 bc	51.7 ± 2.5 de
Gly 1800	75.0 ± 2.6 bc	85.8 ± 2.0 cde	76.7 ± 2.1 b	74.2 ± 3.0 b	57.5 ± 2.1 cd
Gly + Flua	48.3 ± 1.1 fg	80.0 ± 2.9 de	31.7 ± 2.8 f	48.3 ± 2.5 e	39.2 ± 2.0 f
Gly + Ace ^†^	75.8 ± 3.0 bc	85.0 ± 1.8 cde	88.3 ± 2.5 a	87.5 ± 2.5 a	65.8 ± 1.5 c
(Gly + Oxi) + Flua ^†^	53.3 ± 2.1 ef	86.7 ± 2.5 bcd	42.5 ± 1.1 de	36.7 ± 2.1 f	27.5 ± 2.8 g
Gly + Seth	40.8 ± 2.4 g	77.5 ± 1.1 de	45.8 ± 2.7 de	52.5 ± 1.7 de	44.4 ± 1.5 ef
Gly + Cleth	51.7 ± 1.1 ef	75.8 ± 1.5 e	36.7 ± 3.1 ef	49.2 ± 2.4 e	40.8 ± 1.8 f
Glufos	99.2 ± 0.8 a	96.7 ± 1.7 ab	93.3 ± 1.7 a	88.3 ± 1.7 a	79.2 ± 2.0 b
Par + Diu	98.3 ± 1.1 a	85.8 ± 1.5 cde	63.3 ± 2.1 c	57.5 ± 1.7 de	48.3 ± 1.7 def
Glufos + Oxi ^†^	100.0 ± 0 a	99.2 ± 0.8 a	97.5 ± 1.7 a	94.2 ± 2.4 a	90.8 ± 1.5 a
Gly + Oxa ^†^	58.3 ± 2.8 de	76.7 ± 1.7 d	50.8 ± 3.0 d	54.2 ± 2.0 de	45.0 ± 1.8 ef
Gly + (Brom + Diu) ^†^	74.2 ± 2.0 bc	94.2 ± 2.0 abc	95.8 ± 2.4 a	96.7 ± 2.1 a	90.8 ± 2.7 a

^1^ Abbreviations of herbicides: Gly = Glyphosate, Flua = Fluazifop-p-butyl, Ace = Acetochlor, Oxi = Oxifluorfen, Seth = Sethoxydim, Cleth = Clethodim, Glufos = Glufosinate, Par = Paraquat, Diu = Diuron, Oxa = Oxadiazon, Bro = Bromacil. ^†^ Treatments applied in sequence 15 days after the first application. Same letter within a column showed no differences between treatments by the Tukey test (*P* > 0.05). ± Standard error of the mean of two field trials conducted in 2014 and 2015 (*n* = 6).

**Table 4 plants-08-00325-t004:** Cost of herbicide treatments (Cost), relative daily cost considering 75 days of weed control (R-$/day ha^−1^) *, percentage of cost increase (% Inc) * in relation to the representative treatment (RT; 1080 g ae ha^−1^ glyphosate = USD $29.0 ha^−1^), days of weed control <80% (DAT>80%), and cost of satisfactory weed control per day (SC $/day ha^−1^) * in two Persian lime groves of the “San Manuel” Farm, Puebla, Mexico. All costs are estimated in relation to one hectare (1 ha^−1^).

Treatment ^1^	Cost	R-$/day	% Inc	San Manuel I		San Manuel II	
				DAT > 80%	SC $/day	DAT > 80%	SC $/day
Control	-	-	-	-	-	-	-
Gly 1080	29.0	0.38	*RT*	-	-	30	0.96
Gly 1440	38.7	0.51	33	-	-	30	1.29
Gly 1800	48.4	0.65	66	15	3.22	30	1.61
Gly + Flua	100.9	1.35	247	-	-	30	3.36
Gly + Ace ^†^	57.3	0.76	97	30	1.91	60	0.95
(Gly + Oxi) + Flua ^†^	97.4	1.30	183	-	-	30	-
Gly + Seth	82.0	1.09	236	-	-	-	-
Gly + Cleth	70.0	0.93	141	-	-	-	-
Glufos	81.2	1.08	86	60	1.35	75	1.08
Par + Diu	26.8	0.36	−8	30	0.89	30	0.89
Glufos + Oxi ^†^	134.6	1.79	364	75	1.79	75	1.79
Gly + Oxa ^†^	112.0	1.49	286	-	-	-	-
Gly + (Brom + Diu) ^†^	142.4	1.89	391	75	1.89	75	1.89

* R-$/day = cost of a determined treatment/75 days; % Inc = [cost of a determined treatment/$29.0 ha^−1^ (cost of representative treatment)] − 100; and SC $/day = cost/DAT > 80% of a determined treatment. ^1^ Gly = Glyphosate, Flua = Fluazifop-p-butyl, Ace = Acetochlor, Oxi = Oxifluorfen, Seth = Sethoxydim, Cleth = Clethodim, Glufos = Glufosinate, Par = Paraquat, Diu = Diuron, Oxa = Oxadiazon, Bro = Bromacil. ^†^ Treatments applied in sequence 15 days after the first application.

**Table 5 plants-08-00325-t005:** Summary of activities conducted for weed control with pre- (PRE) and post-emergence (POST) herbicides in two Persian lime groves of the “San Manuel” Farm, Puebla, Mexico.

Field Activity	Date	
	2014	2015
Weed mechanical mowing in both experimental plots	16 December 2013	19 December 2014
Initial counting of plants of each weed species	9 January2014	15 January 2015
Application of POST herbicides	10 January2014	16 January 2015
Application of PRE herbicides (treatments 5, 6, 11, 12, and 13), and evaluation at 15 DAT	25 January 2014	31 January 2015
Evaluation at 30 DAT	9 February 2014	15 February 2015
Evaluation at 45 DAT	23 February 2014	1 March 2015
Evaluation at 60 DAT	9 March 2014	15 March 2015
Evaluation at 75 DAT	24 March 2014	29 March 2015

DAT: days after herbicide treatment.

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
