# Peer review of "Management of Glyphosate-Resistant Weeds in Mexican Citrus Groves: Chemical Alternatives and Economic Viability"

_plants, 2019, doi:10.3390/plants8090325_

Round 1
Reviewer 1 Report
The manuscript presented a well designed study of potential glyphosate replacing treatments for weed control in Persian Lime grove fields in Mexico, which is eagerly needed by the citrus growers in the region. The efficacy results can be a guideline to improve the weed management practices in citrus fields in Mexico and other regions. The economic aspect brought up in this manuscript of these alternative treatments is a spotlight in this study, this can give scientific community a different view of the reality of weed management practices adoption.
Although, I recommend that two major items need to be addressed before its considered to be published.
First, the ecological effects of weed community structure that discussed in the article is too far fetched, lack of ecological scaled data to support the statements in this regard. So I suggest to remove the part of implication through out the manuscript.
Second, I strongly suggest the authors to further analyze the cost of treatment into "cost of acceptable weed control/day" to justify the real cost for growers, which will be more tangible for them to guide the adoption/implementation the study findings, which will increase the impact of this study.
Please see the attached file for other comments and corrections.

Author Response
The manuscript presented a well designed study of potential glyphosate replacing treatments for weed control in Persian Lime grove fields in Mexico, which is eagerly needed by the citrus growers in the region. The efficacy results can be a guideline to improve the weed management practices in citrus fields in Mexico and other regions. The economic aspect brought up in this manuscript of these alternative treatments is a spotlight in this study, this can give scientific community a different view of the reality of weed management practices adoption.
Although, I recommend that two major items need to be addressed before its considered to be published.
First, the ecological effects of weed community structure that discussed in the article is too far fetched, lack of ecological scaled data to support the statements in this regard. So I suggest to remove the part of implication through out the manuscript.
Previous lines 204-205 and 256-258 were deleted
Second, I strongly suggest the authors to further analyze the cost of treatment into "cost of acceptable weed control/day" to justify the real cost for growers, which will be more tangible for them to guide the adoption/implementation the study findings, which will increase the impact of this study.
Done: Table 4 was restructured and a new column was added to show the weed control/day
Please see the attached file for other comments and corrections.
The corrections and comments of the attached file were integrated into the manuscript.
Reviewer 2 Report
Abstract – No specific comments.
Introduction – Table of herbicides / herbicide combinations is given in this section but I miss some explanation why these exact herbicides /herbicide combinations were used in the trial – what are proposed benefits of individual treatments?
Results – No specific comments.
Discussion – I miss more detailed discussion about differences between SM-I and SM-II in control efficacy of some treatments to L. virgata.
Materials and Methods – Can authors provide basic data about climate and soil in experimental location?
Conclusions – No specific comments.
Overall comments – English language needs revision.
Author Response
Abstract – No specific comments. OK
Introduction – Table of herbicides / herbicide combinations is given in this section but I miss some explanation why these exact herbicides /herbicide combinations were used in the trial – what are proposed benefits of individual treatments?
The justification for the selected herbicides is in the penultimate paragraph (L70-79) of the introduction
Results – No specific comments. OK
Discussion – I miss more detailed discussion about differences between SM-I and SM-II in control efficacy of some treatments to L. virgata.
The discussion of these results was addressed to highlight those treatments that were efficient in both fields
Materials and Methods – Can authors provide basic data about climate and soil in experimental location?
Requested data were included in the paper in L284-287 and Figure 5.
Conclusions – No specific comments. OK
Overall comments – English language needs revision. English was corrected by a native speaker
Reviewer 3 Report
Line 20 insert to between used and control. Line 34-35, 40-42 is confusing, please reword. There are many sections in the paper that are awkward and confusing and need editing, not just the lines mentioned above.
Author Response
Line 20 insert to between used and control. Done
Line 34-35, 40-42 is confusing, please reword. Done in L32-37 and L42-44
There are many sections in the paper that are awkward and confusing and need editing, not just the lines mentioned above.
Main text was carefully reviewed, long phrases were divided and or rewritten and language was corrected by a native English speaker
Reviewer 4 Report
The presented manuscript compares efficacy of 13 different herbicide treatments on the weeds present in the citrus orchards.
I would recommend to check the draft for proper English and also eliminate typo mistakes (See attachment).
I would also recommend to summarize the output, what would you recommend to the farmer to eliminate the weeds, not only chemical but also other ways of control.

Author Response
The presented manuscript compares efficacy of 13 different herbicide treatments on the weeds present in the citrus orchards.
I would recommend to check the draft for proper English and also eliminate typo mistakes (See attachment). Main text was carefully reviewed and the language was corrected by a native English speaker. In addition, all corrections of the attached file were integrated into the manuscript.
I would also recommend to summarize the output, what would you recommend to the farmer to eliminate the weeds, not only chemical but also other ways of control.
This topic had already been previously addressed in the last paragraph of the discussion and in the conclusion
Round 2
Reviewer 1 Report
The manuscript is improved. However, the "cost per day" to "cost per satisfactory weed control day" need to be addressed correctly, please see comments for details.

Author Response
We appreciate your constructive comments to improve our manuscript.
The "cost per satisfactory weed control day" topic has been addressed as requested.
Sub-section 2.6 was rewritten and table 4 reformulated in the results section (L181-206). The discussion highlighted new points related to the new economic analysis (L264-279). In the methodology a paragraph has been added in L330-334, and the conclusions have been rewritten in the abstract (l34-42) and the main text (L345-351)